# A Fault Detection Method Based on CPSO-Improved KICA

**DOI:** 10.3390/e21070668

**Published:** 2019-07-09

**Authors:** Mingguang Liu, Xiangshun Li, Chuyue Lou, Jin Jiang

**Affiliations:** 1Institute of Industrial Processes Intelligent Control, School of Automation, Wuhan University of Technology, Wuhan 430070, China; 2Department of Electrical and Computer Engineering, Western University, London, ON N6A 5B9, Canada

**Keywords:** fault detection, KICA, the maximum entropy, CPSO

## Abstract

In view of the randomness in the selection of kernel parameters in the traditional kernel independent component analysis (KICA) algorithm, this paper proposes a CPSO-KICA algorithm based on Chaotic Particle Swarm Optimization (CPSO) and KICA. In CPSO-KICA, the maximum entropy of the extracted independent component is first adopted as the fitness function of the PSO algorithm to determine the optimal kernel parameters, then the chaotic algorithm (CO) is used to avoid the local optimum existing in the traditional PSO algorithm. Finally, this proposed algorithm is compared with Weighted KICA (WKICA) and PSO-KICA with Tennessee Eastman Process (TEP) as the benchmark. Simulation results show that the proposed algorithm can determine the optimal kernel parameters and perform better in terms of false alarm rates (FAR), detection latency (DL) and fault detection rates (FDR).

## 1. Introduction

Fault detection and diagnosis play very important roles in the monitoring of industrial processes and have received increasing attention in recent years. Fault detection is the first step to realize final diagnosis of various faults and can give alarms immediately when faults happen at the early stage. Many methods, such as model-based, knowledge-based and data-driven methods, have been proposed to detect faults [1,2]. In the complex modern industrial processes, a lot of data information is gathered through distributed control systems (DCS) and used to train data-driven methods. Thus, the data-driven methods have been widely used in the modern industrial processes, and models and empirical knowledge may not always be required in some cases.

Among data-driven methods, PCA, ICA and PLS have received considerable attention recently. For instance, a dynamic PCA implementation (DPCA) has been used to capture the process dynamics because the system variables are time-correlated [3]. Compared with the traditional PCA method, ICA does better in dealing with non-Gaussian data. In fact, there is no data from industrial processes that follows Gaussian distribution strictly [4,5]. In addition, ICA is superior to PLS as there is no need to determine the input-output relationship of the system, while in some industrial systems the manipulated variables can be regarded as inputs and measurement variables can be regarded as outputs [6]. Therefore, in the actual industrial processes, the ICA method has gradually become a research focus.

Considering the non-linearity of industrial processes, a common solution is to use the kernel method, whose basic idea is to project the original non-linear process data onto a high-dimensional linear feature space. On the basis of the kernel theory, the ICA method has been improved continually to satisfy the needs of various industrial systems.

Peng (2014) proposed a KICA-PCA method, in which KICA was used to isolate the non-Gussian independent information and KPCA was utilized to account for the uncertain parts and extract the principal components [7]. In 2016, Du et al. (2016) presented that some fault data in industrial processes could be separated into normal data and fault data in advance. In order to make use of the data, an improved KICA algorithm, namely FKICA, was proposed [8]. Cai and Tian (2017) held that the KICA method treats every KIC equally and cannot highlight the useful KICs associated with fault information. Consequently, fault information may not be explored effectively, which can result in degraded fault detection performance. To overcome this problem, a new non-linear and non-Gaussian process monitoring method has been proposed with the use of Gaussian mixture model (GMM)-based weighted KICA (WKICA) [9].

However, there is a problem with all the KICA methods mentioned above: most of the current KICA algorithms only rely on experience to determine the size of kernel parameters, but the kernel parameter has a decisive effect on the fault detection effects. As a result, the final detection effect cannot reach the best.

To solve this problem, an algorithm CPSO-KICA based on Chaotic Particle Swarm Optimization (CPSO) and KICA is proposed and compared it with WKICA and PSO-KICA with Tennessee Eastman Process as the benchmark.

The remainder of this paper is organized as follows. In Section 2, the principle of CPSO-KICA for fault detection is introduced, and then the fitness function of CPSO is determined. Subsequently, the proposed CPSO-KICA is implemented on TEP and compared with the methods such as WKICA and PSO-KICA in terms of false alarm rates, detection latency and fault detection rates in Section 3. Section 4 presents the summary and concluding remarks.

## 2. Monitoring Method Using CPSO-KICA

### 2.1. Principle of CPSO-KICA

#### 2.1.1. KICA

In general, the data collected in a factory cannot be linearly separable. Assuming that the data collected by a certain sensor is distributed as shown on the left side of Figure 1, the black line represents the normal data, and the red line represents the noise or other interference signals. Since the traditional ICA algorithm is a linear decomposition method, it cannot separate the noise signal from the normal data in Figure 1 directly.

Kernel function is usually introduced to solve the problem of linear inseparability. The kernel function is to transform linearly inseparable points in the lower dimensional space into linearly separable points in the higher dimensional space. As shown on the right side of Figure 1, after the original data points are converted by the function F(x)=(x-a)(x-b), the data points of different categories can be classified in the two-dimensional space. The data for one of the categories is in the part where F(x) is greater than 0. And the data for the other category is in the part where F(x) is less than 0.

The KICA algorithm mainly comprises two steps: (1) The non-linear process data is projected into high-dimensional feature space, and the whitened data is obtained using the KPCA method; (2) The ICA method is employed to extract KICs from the whitened data.

Suppose that the observed data $x_{i}\in R^{m},1\leq i\leq n$, *m* refers to the number of the variables and *n* represents the number of the samples. The non-linear mapping $\phi \left(\cdot \right):R^{m}\to F$ projects the non-linear data in the original variable space onto the high-dimensional linear feature space *F*. Then the projected data set *F* can be expressed as $\Phi =\left[\phi \left(x_{1}\right)\;\phi \left(x_{2}\right)\;\ldots \;\phi \left(x_{n}\right)\right]\in F\times R^{n}$. Then it is to perform mean centering on Φ to acquire the zero mean feature data, $\bar{\Phi }=\left[\bar{\phi }\left(x_{1}\right)\;\bar{\phi }\left(x_{2}\right)\;\ldots \;\bar{\phi }\left(x_{n}\right)\right]\in F\times R^{n}$. Subsequently, the data of $\bar{\Phi }$ is whitened by KPCA.

The covariance matrix C∈F×F of the data $\bar{\Phi }$ may be estimated by $C=\left(1/n\right)\bar{\Phi }{\bar{\Phi }}^{T}$. However, the functional form of ϕ(·) is unknown, which makes it difficult to obtain the eigenmatrix and eigenvalue of *C* directly. Therefore, the kernel trick is introduced [10]. Define a Gram kernel matrix as $K=\Phi^{T}\Phi$, then the *i*th-row and *j*th-column element can be written as:(1)$${\left[K\right]}_{i,j}=\Phi^{T}\left(x_{i}\right)\Phi \left(x_{i}\right)=k\left(x_{i},x_{j}\right)$$
where $k(x_{i},x_{j})$ is a kernel function. The choice of kernel function k(·,·) determines the non-linear mapping ϕ(·), and we can refer to previous research for the specific selection of kernel function [11]. The Gaussian kernel function is selected here, as shown below:(2)$$k\left(x_{i},x_{j}\right)=exp\left(-\frac{\|x_{i}-x_{j}{\|}^{2}}{c}\right)$$
where *c* denotes the kernel parameter. By using kernel function k(·,·), the inner product of two dimensional feature data in the feature space can be calculated in the input space, without having to know the specific non-linear mapping ϕ(·).

The kernel matrix of zero-mean data $\bar{\Phi }$ can be expressed as:(3)$$\bar{K}={\bar{\Phi }}^{T}\bar{\Phi }=K-1_{n}K-K1_{n}+1_{n}K1_{n}$$
where $1_{n}\in R^{n\times n}$ denotes the matrix whose elements are all equal to 1/n. Let $\lambda_{i}\in R,1\leq i\leq n$ denote the eigenvalues of $\bar{K}$ satisfying the condition $\lambda_{1}\geq \lambda_{2}\geq \cdots \lambda_{n}$, and let $\beta_{i}\in R^{n},1\leq i\leq n$ denote the corresponding eigenvectors of $\bar{K}$. Therefore, the covariance matrix *C* can be expressed as:(4)$$C=\left(\bar{\Phi }H\wedge^{-\frac{1}{2}}\right)\frac{\wedge }{n}{\left(\bar{\Phi }H\wedge^{-\frac{1}{2}}\right)}^{T}$$
where $\wedge =diag\left\{\lambda_{1},\lambda_{2}\ldots \lambda_{n}\right\}\in R^{n\times n},H=\left[\beta_{1}\;\beta_{2}\;\ldots \;\beta_{n}\right]\in R^{n\times n}$. The first α eigenvalues and their corresponding eigenvectors are usually used to calculate the matrix *V* for simplicity:(5)$$V=\bar{\Phi }H_{\alpha }\wedge_{\alpha }^{-\frac{1}{2}}$$
where $\wedge_{\alpha }=diag\left\{\lambda_{1},\lambda_{2}\ldots \lambda_{\alpha }\right\},H_{\alpha }=\left[\beta_{1}\;\beta_{2}\;\ldots \;\beta_{\alpha }\right]$. Therefore, the transformation matrix $Q\in R^{\alpha }\times F$ of the kernel whitened matrix *Z* can be expressed as:(6)$$Q=diag{\left\{\frac{\lambda_{1}}{n},\frac{\lambda_{2}}{n},\ldots ,\frac{\lambda_{\alpha }}{n}\right\}}^{-\frac{1}{2}}V^{T}=\sqrt{n}\wedge_{\alpha }^{-1}H_{\alpha }^{T}{\bar{\Phi }}^{T}$$
Then the kernel whitened data *Z* can be deduced in the feature space as follows:(7)$$Z=\left[z_{1}\;z_{2}\;\ldots \;z_{n}\right]=Q\bar{\Phi }=\sqrt{n}\wedge_{\alpha }^{-1}H_{\alpha }^{T}\bar{K}$$
After the matrix of *Z* is obtained, KICs can be acquired with reference to the literature [12].

However, to implement the KICA algorithm, the kernel parameter *c* needs to be determined first. To demonstrate the role of kernel parameter in fault detection, an example is given with the TE process. TEP is a benchmark to simulate a chemical plant, which is widely used in the field of fault diagnosis to verify the effectiveness of methods. Figure 2 denotes the detection results of KICA under different kernel parameters of Fault 17 in TEP, and the values of kernel parameter *c* are 100, 800 and 1000 respectively, where the red line and blue line represent the size of threshold and statistic respectively. As can be seen from the three figures, different values of the kernel parameter *c* can cause different detection results, but the traditional KICA methods only rely on experience to select the kernel parameters, thus the detection result is not guaranteed to be the best.

Therefore, how to select optimal kernel parameters without experience is particularly important. If the search algorithm is used to select the kernel parameters, the fitness function should be determined first. In this paper, the maximum entropy of the extracted independent component is first adopted as the fitness function to search the optimal *c*.

#### 2.1.2. The Selection of Fitness Function

Empirical methods have always been adopted in KICA based methods to select kernel parameters for different systems. Therefore, a new fitness function is proposed in this paper. And the negative entropy of KIC is taken as the fitness function of the CPSO algorithm to select optimal kernel parameters. Assuming that the first KIC extracted is represented by $y_{1}$, the fitness function can be written as:(8)$$\begin{array}{cc} fitness=max\{f(x)\} & =max\left\{-J\left(y_{1}\right)\right\}\approx -\frac{1}{12}E{\left\{y_{1}^{3}\right\}}^{2}-\frac{1}{48}kurt{\left(y_{1}\right)}^{2}\\ kurt(y_{1}) & =E\left\{y_{1}^{4}\right\}-3{\left(E\left\{y_{1}^{2}\right\}\right)}^{2} \end{array}$$
where f(x) is the fitness function, $J(y_{1})$ and $kurt(y_{1})$ represent the entropy and kurtosis of the variable $y_{1}$.

By contrast, PSO-KPCA takes the maximum eigenvalue as the fitness function. Taking TEP data as an example, this paper compares the usefulness of the two different fitness functions in CPSO-KICA. Figure 3a,b present the graphs of these two functions under different kernel parameters *c* in KICA. As can be seen from Figure 3a, when the eigenvalue is taken as the fitness function, the function is monotonically decreasing. When the kernel parameter *c* is selected according to the maximum eigenvalue, the detection result of CPSO-KICA is similar to Figure 2a, which means that the fault detection result is not the best. Therefore, it is not advisable for CPSO-KICA to take eigenvalue as the fitness function.

The negative entropy is taken as the fitness function, as shown in Figure 3b. Considering three maxima with corresponding *c* values 200, 816 and 1300 respectively. The corresponding detection results are similar to Figure 2a–c, As can be inferred from the comparison of figures, the detection result is better when *c* is equal to 820. Therefore, it is advisable to take negative entropy as the fitness function.

However, the fitness function based on negative entropy is a multi-peak function, the traditional search algorithms such as PSO can easily fall into local optimum. In order to obtain the global optimal solution, the improved CPSO algorithm is used to select kernel parameter *c*.

#### 2.1.3. CPSO

PSO is an optimization algorithm based on the foraging process of birds and shoal. The potential solution to each optimization problem in PSO is a bird in the search space, which is referred to as a particle [13]. Each particle has its own velocity and position, which are *V* and *X* respectively. The fitness value of each particle can be determined by the optimized function, and the velocity and position of the particles are updated according to the Formula (9).
(9)$$\left\{\begin{array}{c} V\left(t+1\right)=wV\left(t\right)+c_{1}r_{1}\left(pbest-X\left(t\right)\right)+c_{2}r_{2}\left(gbest-X\left(t\right)\right)\\ X(t+1)=X(t)+V(t+1) \end{array}\right.$$
where *t* represents the *t*-th iteration, *w* is the weight coefficient of V(t), w∈[0,1], $r_{1}$ and $r_{2}$ are random numbers between [0,1], and $c_{1}$ and $c_{2}$ are learning factors, which affect the speed of particle swarm following the optimal solution. The specific problem-solving process of PSO can be obtained with reference to the literatures [14].

Although PSO has the advantages of fast convergence and few control parameters, it is easy to get into local optimum, especially when the fitness function is a multiple peak function. Therefore, chaos optimization (CO) algorithm is introduced to solve the problem.

Chaos can be defined as the random running state obtained from the determined equation. The CO algorithm is a set of random sequences generated by the chaotic equation through iteration, which are used for random traversal search. Taking advantage of the pseudo-randomness and ergodicity of the variables generated, the chaos algorithm can realize excellent global search function. The Logic mapping is generally used to generate pseudo-random sequences:(10)$$Z:a_{n+1}=\mu a_{n}\left(1-a_{n}\right)$$
where *Z* is a chaotic variable, $a_{n}\in Z\in \left[0,1\right],\mu$ is the control variable. In general, *Z* can be in the full chaotic state when μ=4. Let *Z* have an initial value of $a_{0}\left(a_{0}\neq 0,0.25,0.5,0.75,1\right)$, a sequence is generated by the Formula (10). Suppose $X=[x_{1}\;x_{2}\;\ldots \;x_{k}\in \left[b,c\right]]$ is a variable that needs to be chaotic in PSO algorithm, where *k* represents the number of *x*, namely the number of particles in PSO. Since the value ranges of *X* and *Z* are different, it is necessary to map the variable *Z* generated by the chaotic equation to the range of [b,c]. In this way, the traverse of the value range of *Z* to the variable *X* can be realized: (11)Z→X:X=b+(c-b)Z

After the different fitness values of variable *X* are calculated, the Logic equation should be reintroduced into the area near gbest after each iteration of PSO, so as to realize the local chaotic search:(12)Z→Y:X=gbest+R(Z-0.5)
where *R* is the search radius, which is used to control the search range near gbest. Different from the fixed weight coefficient *w* in PSO, the updating formula of *w* in CPSO is:(13)$$w=w_{min}+\frac{w_{max}-w_{min}}{t}$$
where *t* denotes the number of iterations, and $w_{min}$ and $w_{max}$ denote the lower and upper control limit of *w*. In this way, in the initial stage of iterativeness, the solution search area is increased by increasing the search speed *V*. After constant iteration and stabilization, in the later stage of iteration, CPSO can achieve faster convergence by reducing the search speed *V*.

The flow chart of the CPSO algorithm is shown in Figure 4.

In order to compare the search ability of CPSO and PSO, this section makes a comparison between the optimization results of the two algorithms under a simple multi-peak function, and the equation of the objective function is:(14)$$z=\left(3{\left(1-x\right)}^{2}-19\right)e^{\left(-\left(y^{2}\right)-{\left(x+1\right)}^{2}\right)}-9\left(\frac{x}{5}-x^{3}-y^{5}\right)e^{\left(-x^{2}-y^{2}\right)}$$

As can be seen from Figure 5a, there are two adjacent minimum values of this function, which are −10.6 and −7.8 respectively.

When the PSO and CPSO algorithms are used to find the minimum value of the function respectively, the particle swarm size k=20, the number of iterations t=100, the learning factor $C_{1}=C_{2}=2$, and the maximum speed $V_{max}=2$. The final iteration results of PSO and CPSO are shown in Figure 5b,c, where the blue line represents the optimal fitness value of each iteration. As can be seen from the two figures, the optimal fitness value of the PSO algorithm finally tends to be stable at around −7.8, but CPSO has already converged to −10.6 after about 5 times of iteration.

When the maximum speed of the two algorithms is set to $V_{max}=10$, as shown in Figure 6a,b, the PSO cannot find the minimum value of the function. Because when the speed is too high, its particle swarm always oscillate around the minimum value. Consequently, the optimal adaptive value cannot converge. However, the CPSO algorithm can still find the optimal value quickly.

It can be concluded from the comparison that: PSO may easily get trapped in a local optimum, and it is difficult to converge to optimal fitness value, when the $V_{max}$ is too high. If a better fitness value needs to be obtained, some parameters need to be selected by experience, such as the maximum speed $V_{max}$. In contrast, CPSO not only can find the minimum value, but also has fast convergence speed, with no strict requirements on parameters. Therefore, the CPSO algorithm has stronger search ability and robustness than the PSO algorithm.

Hence, taking negative entropy as the fitness function, this paper employes the CPSO algorithm to find the optimal kernel parameter *c*. The simulation result is shown in Figure 7. It can be seen from this figure that CPSO can quickly find the maximum negative entropy, and the correspond kernel parameter c=820.

After the kernel parameter *c* of different types of data is determined with CPSO, the system can be modelled and detected with the KICA algorithm.

### 2.2. The Process Monitoring Methods

Support Vector Data Description (SVDD) is introduced for process monitoring here. Suppose $X=\{x_{i},i=1,2,\ldots ,n\}$, with *n* being the number of samples. The non-linear data in the original variable space is projected onto the high-dimensional linear feature space *F* by using the non-linear mapping ϕ(·), the mathematical idea of SVDD is to use the smallest hypersphere to envelope the normal data in the feature space *F*,and it is to solve the optimization problem:(15)$$\begin{array}{c} min\left(R^{2}+C\sum_{i=1}^{n}\varepsilon_{i}\right),\varepsilon_{i}>0\\ s.t.\|\phi (x_{i}){-\alpha \|}^{2}\leq R^{2}+\varepsilon_{i},\varepsilon_{i}\geq 0,i=1,2,\ldots ,n \end{array}$$
where α is the center of the hypersphere, *R* is the radius of the hypersphere, and $\varepsilon_{i}$ is the relaxation variable. The specific steps for solving this quadratic programming problem can be obtained with reference to previous studies [15,16]. Then the radius *R* of the hypersphere can be acquired:(16)$$R^{2}={\left\|\phi \left(x_{s}\right)-\alpha \right\|}^{2}=1-2\sum_{i=1}^{n}a_{i}K\left(x_{s},x_{i}\right)-\sum_{i=1}^{n}\sum_{j=1}^{n}K\left(x_{i},x_{j}\right)$$
where, $x_{s}$ represent support vectors, which are the points in the hypersphere. When the new detection data $x_{new}$ are obtained, the new distances *D* between $x_{new}$ and α are calculated as:(17)$$D^{2}={\left\|\phi \left(x_{new}\right)-\alpha \right\|}^{2}=1-2\sum_{i=1}^{n}a_{i}K\left(x_{new},x_{i}\right)-\sum_{i=1}^{n}\sum_{j=1}^{n}K\left(x_{i},x_{j}\right)$$
Then, Formula(18) can be used to judge whether the new data $x_{new}$ is normal or not:(18)$$\left\{\begin{array}{c} D^{2}\leq R^{2}\Rightarrow \mathrm{fault}\;\mathrm{free}\\ D^{2}>R^{2}\Rightarrow \mathrm{faulty} \end{array}\right.$$

Moreover, fault diagnosis should be achieved after fault detection. Methods and applications of fault diagnosis are introduced in some references [17,18,19]. The method of cumulative contribution histogram has been applied here to isolate the fault variable. Once the fault is detected, the time indices the fault start and end is determined, marked as $i_{s}$ and $i_{e}$. The contribution value of the *j*-th variable to the fault at time $i\in [i_{s},i_{e}]$ can be expressed as: (19)$$cv\left(i,j\right)={\left(y_{new(i,j)}-y_{\left(i,j\right)}\right)}^{2}$$
where $y_{new}$ and *y* represent KICs extracted from the new test data and normal trained data respectively. Thus, the contribution value of the *j*-th variable to the fault in the whole time interval of fault is written as:(20)$$CV\left(j\right)=\sum_{i}cv\left(i,j\right)$$

Thus, the corresponding variable with the largest CV is the fault source.

## 3. Simulation

In order to verify the detection ability of CPSO-KICA, TE process is applied for fault detection. Through a comparison FAR, FDR and DL with WKICA [9] and PSO-KICA, the practicability of the proposed method is judged.

### 3.1. Data Acquisition

Figure 8 presents the diagram of the Tennessee Eastman Process. The plant consists of five major units: reactor, stripper, separator, condenser and compressor [20].

TEP is a typical chemical process which can generate large amounts of data. The process has 22 continuous process measurements, 19 composition measurements and 12 manipulated variables [21]. Because three of these manipulated variables are constant, only the other nine manipulated variables are selected, as listed in Table 1. There are 20 different types of faults inserted in the system. Among the faults, IDV (6) is the only type of destructive fault that can cause the system to shut-down.

The simulation data is obtained on the TEP simulation platform, and a total of 960 samples are collected. The corresponding fault is introduced at the 161th sample with a sampling interval of 1 s. The normal data is used as training data to obtain the threshold of statistics. And the fault data is used as monitoring data to test the superiority of the proposed method. Because IDV(6) leads to system shut-down, only 550 samples of data are collected, in which the first 500 samples are normal data. In order to eliminate the noise in the collected data, this paper carries out the wavelet denoising before data preprocessing. Some articles about the wavelet denoising can be read for Reference [22].

### 3.2. Results of the Experiment

Three detection indexes are used in order to verify the superiority of the improved algorithm, which can be expressed as:(21)$$\begin{array}{cc} FDR & =\frac{n}{N}\times 100\\ DL & =L-l\\ FAR & =\frac{d}{D}\times 100 \end{array}$$
in the calculation of the FDR, *N* is the total number of faulty samples within the interval of analysis, and *n* is the number of correctly detected faulty samples. In the calculation of the DL, *l* is the *l*th sample point where the fault is detected for the first time, and *L* represents the sequence number of the sample points that has a fault for the first time. And in the calculation of the FAR, *d* is the number of normal samples incorrectly classified as faulty samples and *D* is the total number of normal samples within the interval of analysis. The results of FDR are summarized in Table 2. The optimal value for each failure has been highlighted in bold font.

In order to compare the detection rates of these three methods more intuitively, a line chart is utilized to display their detection rates, which can be seen in Figure 9. The blue, red and black lines represent the detection rates of WKICA, PSO-KICA and CPSO-KICA respectively.

It can be seen from the blue and black lines in Figure 9 that the detection rate of CPSO-KICA is obviously better than that of PSO-KICA, which further verifies that CPSO is better than the traditional PSO algorithm in finding the optimal kernel parameter. In addition, it can be seen from the blue line and the red line that the proposed CPSO-KICA-based method exhibits higher fault detection rates than WKICA for most faults except faults 5, 12 and 16, and that the detection rates for most faults are about 99%. Although the faults 3,9 and 15 have been testified to be extremely difficult to detect for the data driven monitoring methods due to that there are no observable changes in the mean or the variance [23], faults 3 and 9 can still be detected.

FDR, FAR and DL usually are used to measure the efficiency of an algorithm for fault detection. The objective of fault detection methods is to achieve the highest FDR, the lowest FAR and DL. The detection results of DL and FAR are presented in Table 3. It can be seen from the table that CPSO-KICA can detect faults quickly with zero FAR and low DL.

For example, fault 3 is caused by the step change in the D feed temperature, and the change of this temperature is shown in Figure 10. However, as a matter of fact, the average temperature of the reactor outlet changes from 102.48 to 102.44, which is very small. In addition, the system has a closed loop control, and other variables barely change, so many data-based methods could not detect the fault.

The monitoring results under fault 3 are given in Figure 11 and Figure 12. As can be seen from the two figures, the WKICA is not able to detect fault 3 and provides a false alarm before the fault occurs. While CPSO-KICA method can detect the fault without false alarm. The reason is that CPSO is able to find the optimal kernel parameter *c*, which maximizes the negative entropy of the extracted KICs, thus extracting more useful information. In addition, SVDD is used to detect abnormal data to improve the FDR.

Figure 13 shows the cumulative contribution graph of 50 variables in fault 3. As can be seen from the figure, the value of variable 21 is the largest, and variable 21 corresponds to the outlet temperature of the reactor. And the reason for fault 3 is the change of feed temperature of D. Therefore, this method has the ability of fault diagnosis for TEP.

## 4. Conclusions

In this paper, the CPSO-KICA algorithm is proposed to solve the problem of difficult selection of kernel parameters in traditional KICA algorithm. The maximum negative entropy of KIC extracted is taken as the fitness function of CPSO. It can not only find the optimal kernel parameter, but also avoid the local optimum. To overcome the shortcomings of $I^{2}$ and *Q* statistics, the present study utilizes the SVDD to detect the abnormal data. Finally, CPSO-KICA is compared with PSO-KICA and WKICA by using the data collected from TEP. The results show that CPSO-KICA is very efficient in fault detection applications with higher fault detection rates and lower detection latency. Moreover, there is no false alarm for different types of faults.

As a concluding remark, it should be pointed out that the determination of fault identification is still an open problem. Although the fault can be detected efficiently by CPSO-KICA, there is still no good method to identify the fault source. And the identification of RUL (remaining useful life) is usually the ultimate goal of such algorithms, but the method in this paper can not meet the corresponding requirements. In addition, the Logistic map suffers from some common weaknesses in CPSO such as a stable windows, a relatively small key space and an uneven distribution of sequences; these problems could also be the focus of future research.

## Figures and Tables

**Figure 1 entropy-21-00668-f001:**
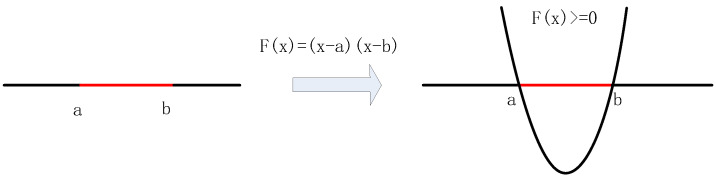
Linearly non-separable and linearly separable data.

**Figure 2 entropy-21-00668-f002:**
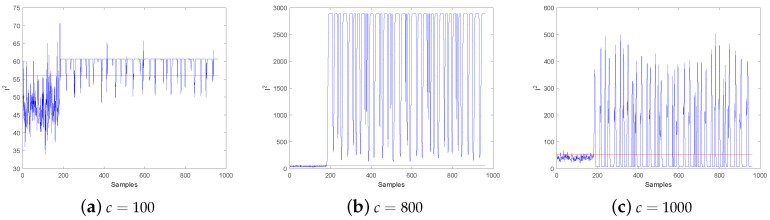
Detection Results of Fault 17 with Different kernel Parameters *c*.

**Figure 3 entropy-21-00668-f003:**
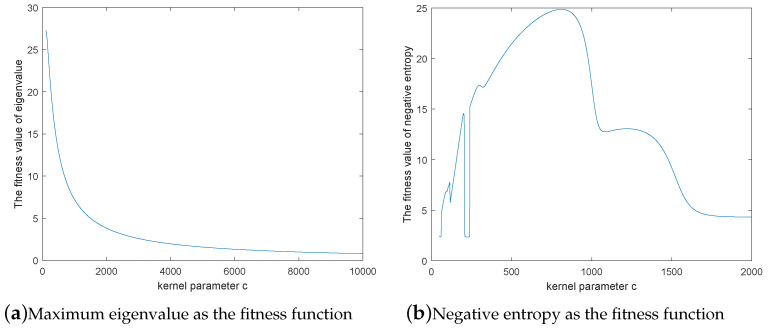
Comparison of two fitness functions

**Figure 4 entropy-21-00668-f004:**
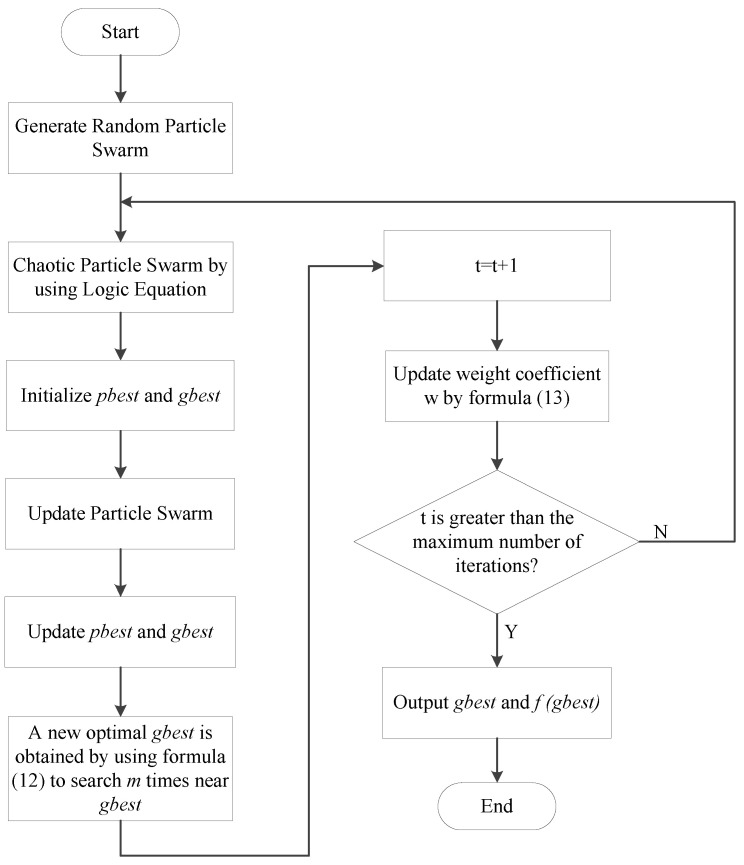
Flow chart of the CPSO algorithm.

**Figure 5 entropy-21-00668-f005:**
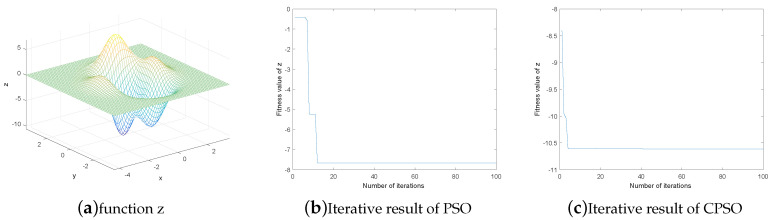
Comparison of iteration results.

**Figure 6 entropy-21-00668-f006:**
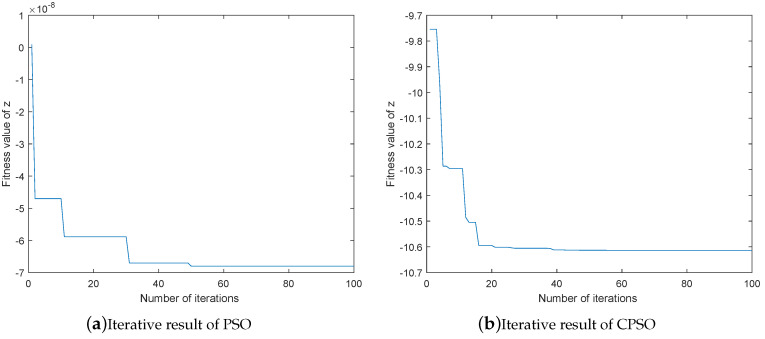
Comparison of iteration results.

**Figure 7 entropy-21-00668-f007:**
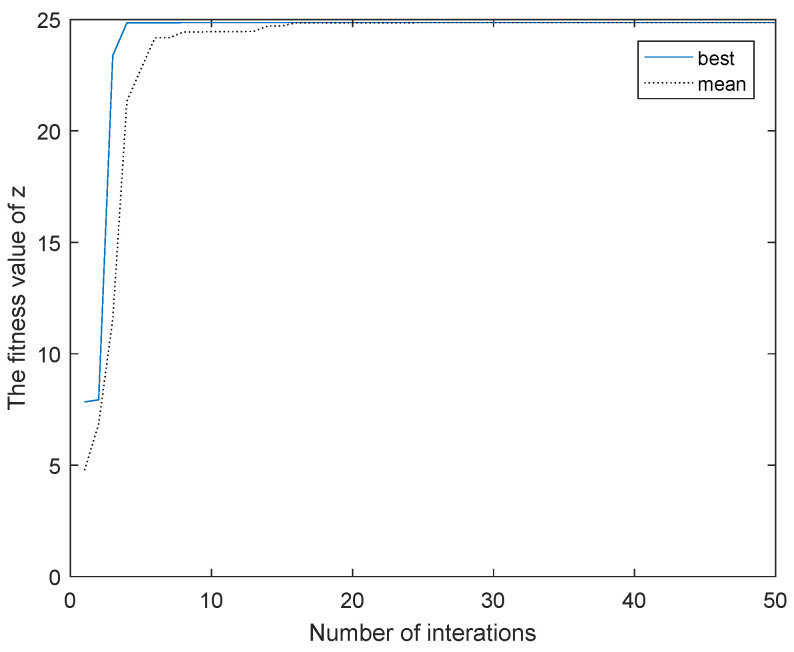
Result of CPSO.

**Figure 8 entropy-21-00668-f008:**
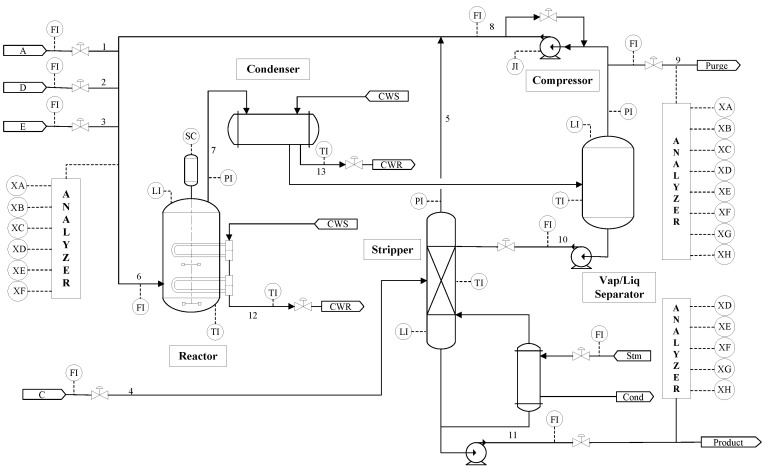
Tennessee Eastman process.

**Figure 9 entropy-21-00668-f009:**
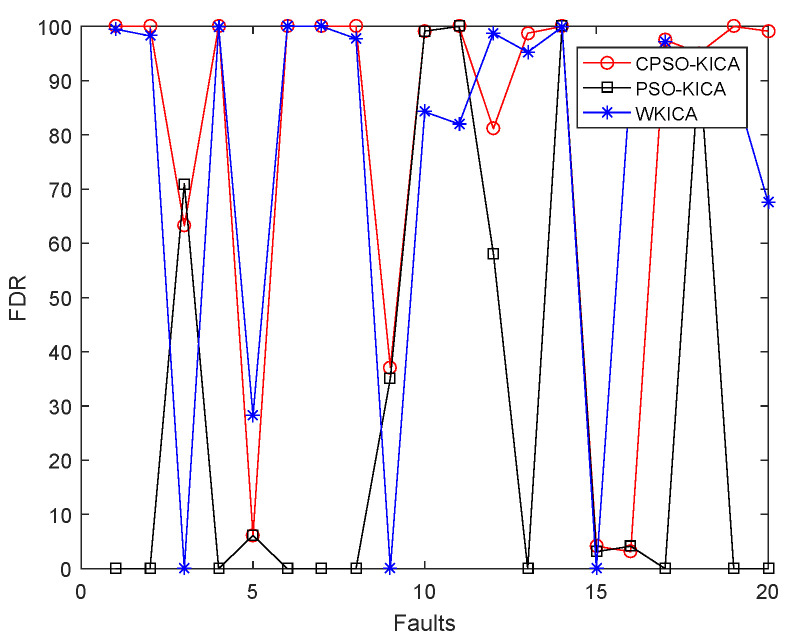
Recognition accuracy.

**Figure 10 entropy-21-00668-f010:**
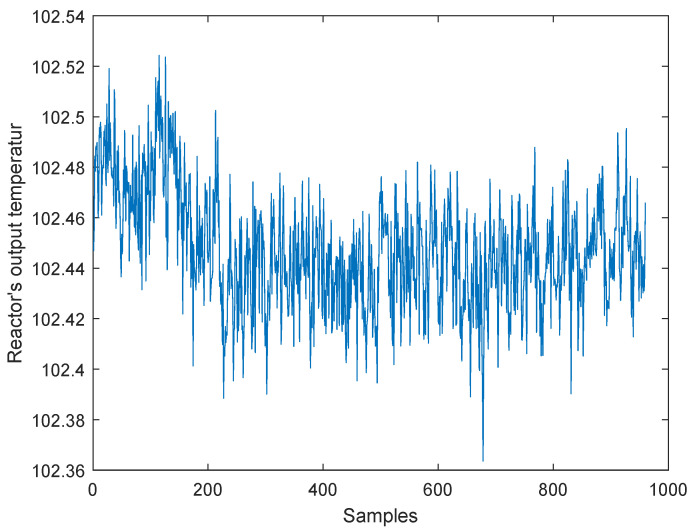
D feed temperature.

**Figure 11 entropy-21-00668-f011:**
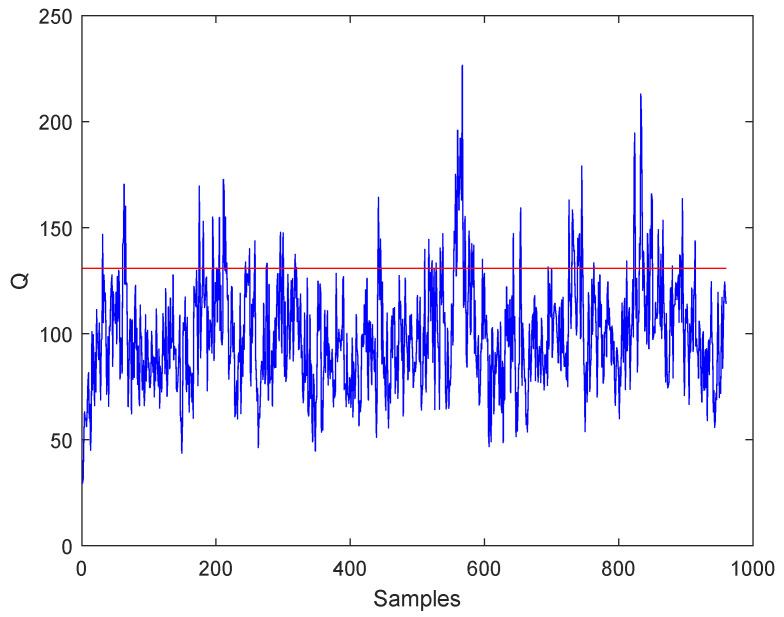
Fault 3:Detection results of WKICA.

**Figure 12 entropy-21-00668-f012:**
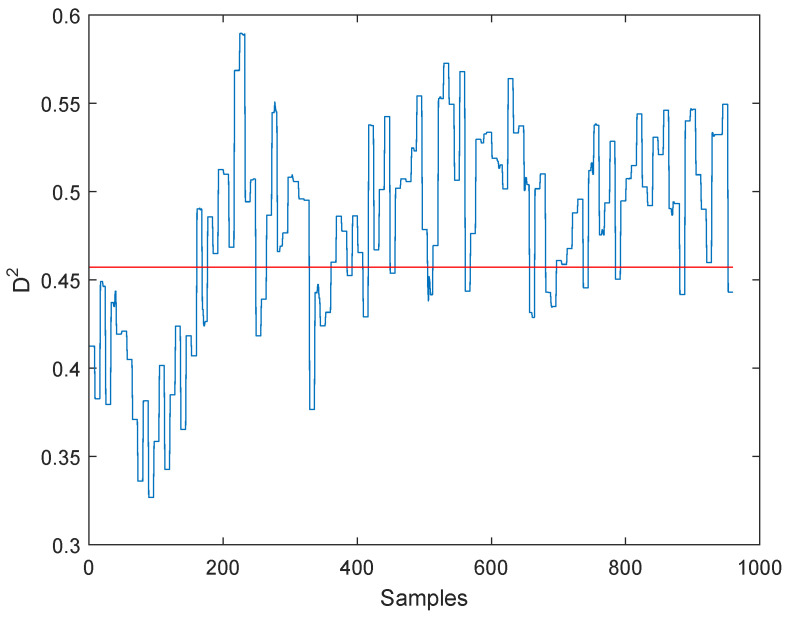
Fault 3:Detection results of CPSO-KICA.

**Figure 13 entropy-21-00668-f013:**
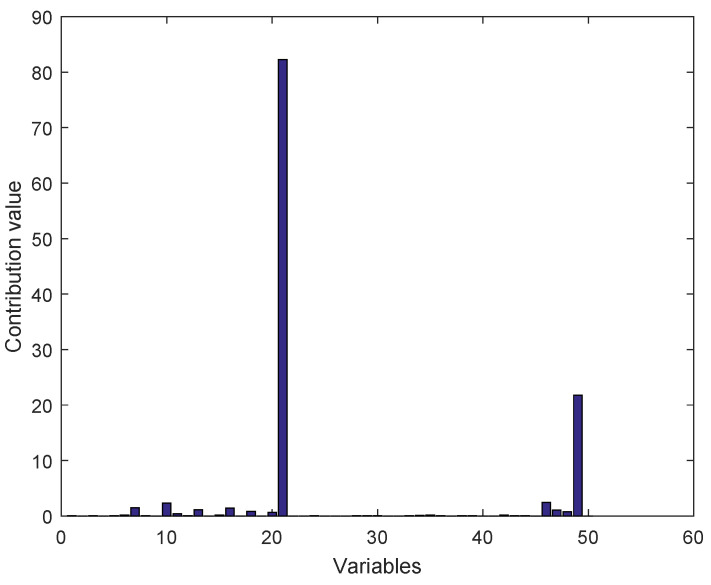
Cumulative contribution diagram of fault 3.

**Table 1 entropy-21-00668-t001:** Manipulated Variables.

Variables	Description
XMV(42)	D feed flow (stream 2)
XMV(43)	E feed flow (stream 3)
XMV(44)	A feed flow (stream 1)
XMV(45)	A and C feed flow (stream 4)
XMV(46)	Purge valve (stream 9)
XMV(47)	Separator pot liquid flow (stream 10)
XMV(48)	Stripper liquid product flow (stream 11)
XMV(49)	Reactor cooling water valve
XMV(50)	Condenser cooling water flow

**Table 2 entropy-21-00668-t002:** TEP: fault detection rate (%).

Fault	WKICA		PSO-KICA	CPSO-KICA
	Q	$I^{2}$	$D^{2}\left(c=200\right)$	$D^{2}\left(c=820\right)$
1	99.50	99.38	0.00	**100.00**
2	98.25	98.38	0.00	**100.00**
3	failed	failed	**71.00**	63.25
4	99.88	91.62	0.00	**100.00**
5	**28.25**	25.62	6.13	6.13
6	**100.00**	**100.00**	0.00	**100.00**
7	**100.00**	99.88	0.00	**100.00**
8	97.75	97.38	0.00	**100.00**
9	failed	failed	35.13	**37.13**
10	84.25	79.12	**99.13**	**99.13**
11	82.00	50.88	**100.00**	**100.00**
12	98.75	**99.25**	58.13	82.13
13	95.25	94.75	0.00	**98.75**
14	99.88	99.75	**100.00**	**100.00**
15	failed	failed	3.13	**4.13**
16	**90.38**	78.50	4.13	3.13
17	97.25	91.62	0.00	**97.50**
18	90.50	89.62	93.13	**94.13**
19	89.38	54.25	0.00	**100.00**
20	67.62	54.75	0.00	**99.13**

**Table 3 entropy-21-00668-t003:** TEP: fault detection rate (%).

Fault	WKICA		CPSO-KICA	
	DL(Q)	FAR	DL	FAR
1	5	1.67%	**0**	**0%**
2	17	1.67%	**0**	**0%**
3	failed	1.67%	**0**	**0%**
4	2	1.67%	**0**	**0%**
5	1	1.67%	56	**0%**
6	**0**	1.67%	**0**	**0%**
7	**0**	1.67%	**0**	**0%**
8	19	1.67%	**0**	**0%**
9	failed	1.67%	**16**	**0%**
10	28	1.67%	**7**	**0%**
11	10	1.67%	**0**	**0%**
12	**2**	1.67%	7	**0%**
13	47	1.67%	**5**	**0%**
14	1	1.67%	**0**	**0%**
15	failed	1.67%	**696**	**0%**
16	10	1.67%	696	**0%**
17	21	1.67%	**5**	**0%**
18	81	1.67%	*39*	**0%**
19	10	1.67%	**0**	**0%**
20	80	1.67%	**8**	**0%**

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
