# Peer review of "A Fault Detection Method Based on CPSO-Improved KICA"

_entropy, 2019, doi:10.3390/e21070668_

Round 1

Reviewer 1 Report

The application described in the paper is interesting, but in some parts the description of the methodology is confusing or incomplete, which would hinder its reproduction. Here are some observations and suggestions:

The following paper can be added to improve the introduction and the problem statement in order to compare with other methods based on data-driven fault detection:

https://doi.org/10.1016/j.ifacol.2018.09.604

https://doi.org/10.1016/j.conengprac.2018.09.006

2. It is not clearly specified what has been plotted in the graphs of Figure 2. How to interpret these graphs in the context of fault detection?

3. What do f(x) and J(y) represent in equation (8)? This should be specified or indicate a bibliographic reference where this notation is explained. Remember to define each variable or notation before using it for the first time.

4. The third paragraph of section 2.1.3 refers to the use of equation (8) to generate the pseudo-random sequence {a_n}. How is this possible if equation (8) does not refer to any sequence?

5. The use of the variable "V" (search speed, section 2.1.3), the variable "pbest" and the coefficient "W" (flow diagram in Figure 4) is not clear, so the explanation becomes confusing. Is there any reference where that notation is explained?

6. For clarity, you should use plain text (upright) for embedded text within the equations; e.g., "fault free" in equation (17).

7. Have you tried the proposed methodology with any real physical system or with a different benchmark than the TEP? Is it possible to generalize the results for the detection of faults in other processes?

Author Response

First, I gratefully acknowledge the Reviewer for their valuable and constructive comments. All the issues that the respected reviewer raised has been carefully addressed in the following. I have exerted my best effort to consider these comments that have improved the quality of the results. Hopefully, the revised version is to your satisfaction. Please refer to the uploaded word for specific content.

Reviewer 2 Report

Authors do present an interesting approach to fault identification and classification. Computational results are used to compare the functionality of the proposed system to other existing classifiers. However, the manuscript should be revised before it could be recommended for publication. 

#1. The overview of the existing techniques and algorithms used for early fault diagnosis is too short. Authors should expand the literature overview and include some major techniques developed during the last decade. 

#2. Authors discuss the problems related to the identification of faults. However, this is only one part of the story. The identification of RUL (remaining useful life) is usually the ultimate goal of such algorithms (fault identification being just the first step). An appropriate discussion is required on RUL.

#3. The presentation style in some parts of the manuscript is in fact rather primitive. A typicl example is the extension of PSO to CPSO. Authors introduce the Logistic map and declare that it generates the full chaotic state at miu = 4. However, it is well known that the Logistic map suffers from some common weaknesses such as stable windows, a relatively small key space and an uneven distribution of sequences. These problems can be solved by utilising the intertwined Logistic map. Authors are not necessarily required to modify their algorithm - but at least some discussions on these issues are required. A typical reference could be: The rank of a sequence as an indicator of chaos in discrete nonlinear dynamical systems. Communications in Nonlinear Science and Numerical Simulation (2011) vol.16, p.2894-2906.

#4. The selection of the Kerne functions also looks primitive (Fig. 1). Why authors do not use stochastic variables - this would be a natural choice for data contaminated by unpredictable noise. Moreover, why authors do consider a single continuous and convex set (a simple interval in 1D)? Much more complex kernels would be required if the set topology is more complex. A discussion would be required on these issues too. 

Author Response

(The authors gave the same response as above.)

Reviewer 3 Report

The submitted article deals with the detection of faults in industrial machinery and processes based on the application of CPSO improved KICA. The article deals with a topic usually interesting for researchers and professionals, based on the wide application of industrial machines/processes. The manuscript design was in good academic style and there was a good balance between applicability and theoretical background information. This was clearly demonstrated in Section 2. 

The authors stated that it was impossible to characterise machine conditions and processes during real industrial operating conditions due to the presence of noise which may cause some non-linearities. The authors also stated that liner classification techniques such as PCA are not appropriate. However, I will urge the authors to also consult articles that have used some of these linear techniques for classification of various rotating machines conditions, processes and buildings that were highly susceptible to noise, using data fusion and coherence for noise suppression to values that allowed for the recognition of faults features. Some of such articles that I suggest you consult and reference include:

 Nembhard AD, Sinha JK, Yunusa-Kaltungo A. Development of a generic rotating machinery fault diagnosis approach insensitive to machine speed and support type. Journal of Sound and Vibration. 2015 Feb 17;337:321-41.

Li S, Wen J. A model-based fault detection and diagnostic methodology based on PCA method and wavelet transform. Energy and Buildings. 2014 Jan 1;68:63-71.

Yunusa-Kaltungo A, Sinha JK, Nembhard AD. A novel fault diagnosis technique for enhancing maintenance and reliability of rotating machines. Structural Health Monitoring. 2015 Nov;14(6):604-21.

Ding S, Zhang P, Ding E, Naik A, Deng P, Gui W. On the application of PCA technique to fault diagnosis. Tsinghua Science and Technology. 2010 Apr;15(2):138-44.

Yunusa-Kaltungo A, Sinha JK. Sensitivity analysis of higher order coherent spectra in machine faults diagnosis. Structural Health Monitoring. 2016 Sep;15(5):555-67.

Shams MB, Budman HM, Duever TA. Fault detection, identification and diagnosis using CUSUM based PCA. Chemical Engineering Science. 2011 Oct 15;66(20):4488-98.

Luwei KC, Yunusa-Kaltungo A, Sha’aban YA. Integrated Fault Detection Framework for Classifying Rotating Machine Faults Using Frequency Domain Data Fusion and Artificial Neural Networks. Machines. 2018 Nov 20;6(4):59.

If these minor issues are adequately addressed, I am happy to recommend the manuscript for publication.

Author Response

(The authors gave the same response as above.)

Round 2

Reviewer 1 Report

The authors have answered all of my concerns. I don't have more remarks. 

Reviewer 2 Report

Authors have performed an appropriate revision and the manuscript can be recommended for publication.